# Back Pain without Disease or Substantial Injury in Children and Adolescents: A Twin Family Study Investigating Genetic Influence and Associations

**DOI:** 10.3390/children10020375

**Published:** 2023-02-14

**Authors:** Tessa Beerstra, Minh Bui, Tiina Jaaniste, Aneeka Bott, John Hopper, G. David Champion

**Affiliations:** 1Faculty of Medicine, University of Groningen, 9712 CP Groningen, The Netherlands; 2Centre for Epidemiology and Biostatistics, University of Melbourne, Melbourne, VIC 3010, Australia; 3Department of Pain, Sydney Children’s Hospital, Bright Alliance Building, High St, Randwick, NSW 2031, Australia; 4School of Clinical Medicine, University of NSW, Kensington, NSW 2052, Australia; 5Aneeka Bott Psychology, 69 Arthur St, Randwick, NSW 2031, Australia

**Keywords:** back pain, primary pain, twin family study, pediatrics, genetic, questionnaire survey

## Abstract

This twin family study first aimed to investigate the evidence for genetic factors predicting the risk of lifetime prevalence of non-specific low back pain of at least three months duration (LBP (life)) and one-month current prevalence of thoracolumbar back pain (TLBP (current)) using a study of children, adolescents, and their first-degree relatives. Secondly, the study aimed to identify associations between pain in the back with pain in other regions and also with other conditions of interest. Randomly selected families (*n* = 2479) with child or adolescent twin pairs and their biological parents and first siblings were approached by Twins Research Australia. There were 651 complete twin pairs aged 6–20 years (response 26%). Casewise concordance, correlation, and odds ratios were compared for monozygous (MZ) and dizygous (DZ) pairs to enable inference about the potential existence of genetic vulnerability. Multivariable random effects logistic regression was used to estimate associations between LBP (life) or TLBP (current) as an outcome with the potentially relevant condition as predictors. The MZ pairs were more similar than the DZ pairs for each of the back pain conditions (all *p* values < 0.02). Both back pain conditions were associated with pain in multiple sites and with primary pain and other conditions using the combined twin and sibling sample (*n* = 1382). Data were consistent with the existence of genetic influences on the pain measures under the equal environments assumption of the classic twin model and associations with both categories of back pain were consistent with primary pain conditions and syndromes of childhood and adolescence which has research and clinical implications.

## 1. Introduction

Back pain in children and adolescents without substantial trauma or disease is prevalent and is the largest category of back pain in this age group [1,2]. The reported lifetime prevalence of low back pain (LBP) in adolescents ranges from 10% to 70% in previous studies [3,4,5], and prevalence rates increase with age across adolescence for both males and females [3,4,5,6]. Juvenile back pain is a strong predictor of back pain in adulthood, and there also appears to be a relationship with the prevalence of other common chronic or recurrent pediatric pain conditions, resulting in higher disability [7].

Descriptive diagnostic terminology has included non-specific, functional or functional somatic, idiopathic, and medically unexplained back pain. In the new diagnostic classification of the International Association for the Study of Pain (IASP) for International Classification of Diseases-11 (ICD-11), the recommended term is chronic primary musculoskeletal (thoracic back or low back) pain or pain syndrome [8]. The ICD-11 diagnostic classification is somewhat controversial [9,10,11] and has not yet been widely applied. It has only recently been mentioned in pediatric chronic pain [12,13,14] and was not mentioned in the comprehensive review by Frosch et al. [1], in which the term non-specific back pain was applied.

Chronic primary pain is defined [8] as pain in one or more anatomical regions that (1) persists or recurs for longer than three months; (2) is associated with significant emotional distress (e.g., anxiety, anger, frustration, or depressed mood) and/or significant functional disability (interference in activities of daily life and participation in social roles); and (3) the symptoms are not better accounted for by another diagnosis. The implication is that the biopsychosocial paradigm for assessment and management is appropriate [15]. Chronic primary pain is distinct from chronic secondary musculoskeletal pain with underlying disease and pathology and from chronic post-traumatic pain, where trauma has been clearly dominant in causation. As to the underlying neurobiological mechanism, chronic primary back pain is not neuropathic, and individual back pain conditions usually involve nociceptive central nervous system mechanisms, commonly with nociplastic features [16].

In a twin family publication on the common primary pain disorders of childhood [14], we summarized the characteristics of the common primary pain conditions of childhood (including migraine, non-migraine headaches, recurrent abdominal pain, and growing pains): spontaneous or easily provoked onset, female preponderance, no biomarkers or pathology identifiable by routine clinical or imaging investigations, disordered somatosensory processing in the central nervous system, multiple comorbidities or associations with each other, and with additional disorders including anxiety and depression, and medical conditions such as asthma. They also tend to have multiple risk markers (associations that might be causal), risk of later-life adverse pain, and psychological outcomes, and wide familial and community prevalence.

In this twin family research study, now focusing on back pain, we aimed to investigate familial and genetic influence by applying the classical twin family design. This design stipulates that a greater similarity of MZ twins, who share 100% genes, than the DZ twins who share on average 50% genes, is evidence in favor of familial influence (genetic and/or shared environment), assuming MZ and DZ twins to have equal shared environmental factors. As applied in Champion et al. [14], this design is also applicable to the determination of associations [17] with primary low back and thoracic back pain.

Genetic influence on primary back pain in children and adolescents is to be expected in view of the twin study and genomic evidence for back pain in adults [18,19,20,21,22,23]. Furthermore, there is evidence for the familial occurrence of back pain and parental transmission [24]. Parental transmission of pain generally involves genetic and environmental influences such as parental modeling and reinforcement, stress, injury, and illness [25].

The presence of multiple pain sites in the mother consistently predicts the presence of multiple pain sites in the child, with a site-specific relationship between back pain and headache [26]. However, unexpectedly, the one-twin study of back pain in children showed no evidence for the heritability of back pain [27].

There have been many studies of associations with back pain, mostly in adults and notably showing lifestyle associations, but our focus is more narrowly on the associations which we found to be characteristic of primary pain conditions and syndromes in children and adolescents [17]. These associations were other primary pain conditions, restless legs syndrome, history of iron deficiency, and anxiety and depression. For this study, we have included a measure of multiple sensory sensitivity and have added to the association analyses of anatomical regions of pain.

In addition to genetic causal influences, the identified associations include potentially causal risk factors, which may guide further research and be useful in optimal clinical management.

The aims, in summary, are:To investigate the heritability of chronic low back pain and one-month thoracolumbar back pain in children and adolescents using the classical twin family design;To identify associations with the back pain conditions which are most relevant to primary pediatric pain conditions;Thereby, in chronic low back pain without disease or substantial injury, to confirm or otherwise that it is classifiable as a chronic primary pain syndrome.

## 2. Materials and Methods

### 2.1. Design

A cross-sectional twin family study was used [28], with the inclusion of parents and siblings contributing to the assessment of heritability and associations.

### 2.2. Recruitment, Participants, and Procedure

Questionnaires were sent in two rounds and were distributed by Twins Research Australia (TRA), which is a national volunteer resource of twin pairs and higher-order multiples willing to consider participating in health, medical, and scientific research [28]. Further information about the TRA can be found at https://www.twins.org.au/research/twin-and-data-resource/70-membership (accessed on 12 January 2023). In total, we approached 2479 randomly selected twin pairs aged 11–20 years and their biological parents and siblings. The wide child and adolescent age range selected for this study was determined by considering the differing peak ages of incidence and prevalence of the primary pain conditions and other conditions of interest.

Families who were approached in the first round were also involved in previous phases of our twin studies and had completed questionnaires primarily about other pain conditions of childhood. In the second round, questionnaires were sent to 1102 new families not previously registered with the TRA. Participants could choose to complete the questionnaires on paper or online. The questionnaires were designed to assess LBP and other pain regions. The questionnaires are provided in Appendix A. The first section contained general questions about gender and age. In the second part, all participants were asked if they had experienced pain in any parts of their body for most of the last month and whether their pain was due to injury. To define the pain location, participants were asked to mark all pain sites on the SUPER-KIDZ body map where they had experienced pain (Figure 1) [29]. In the case of more than one affected site, they were asked to highlight the most important pain area. In the online version of the questionnaire, participants had to click twice on the same area to mark it as most important. They were then asked whether they had ever experienced LBP, which lasted for at least three months. To define the location of LBP we added another body map with a specific shaded area, enquiring about any pain in the shaded area. If lifetime prevalence of LBP lasting at least three months was present, participants were requested to answer an additional six questions about the cause (especially whether or not due to injury), whether the pain had ever spread down the leg, the age of onset, whether their daily pursuits were limited (school, sports), and the frequency of the back pain over the last six months (every day, more than once a week, every week, every month, or rarely/never).

The body map consisted of 21 body sites. The sites marked by participants were subdivided into nine regions: lower limbs, upper limbs, chest, abdomen, head, neck, lower back, middle back, and upper back. Thoracolumbar back pain (TLBP) was defined as pain in the lower and/or middle back. Lower limbs contained the right and left foot, ankle, shin, calf, thigh, and hip.

The carers of the twins were also asked to complete a validated zygosity questionnaire to define whether the twins were monozygous (MZ) or dizygous (DZ) [30]. Most zygosities were determined from TRA records of previous DNA tests. The remaining twins’ zygosities were confirmed by information from the validated zygosity questionnaire, which is known to have a 95% accuracy rate [30,31].

### 2.3. Measures and Diagnostic Categorization

The questionnaire content is summarized in Table 1.

Low back pain (LBP (life)). Respondents were classified as having LBP if they responded positively to the question, “During your lifetime ‘Have you’ or ‘Has your child’ ever had pain in their low back in the shaded area, which lasted for at least three months?” Participants received the body map with shading in the area between the lower ribs and the lower gluteal folds to assist with recognizing the subject area. If this question was answered positively, further information regarding the cause of the pain (injury, or without obvious reason), the age of onset, whether the pain spread down the leg, if the pain limited daily activities, and the frequency of LBP in the past six months (daily, weekly, monthly). Individuals were classified as cases for chronic LBP (life) if they experienced the condition for at least three months during their lifetime. Cases of disease-related or severe injury-related back pain were excluded.

Thoracolumbar back pain (TLBP (current)) and other regions of current pain were identified by positive responses to the question “Have you had pain for most of the last month in any parts of your body?” followed by “If you answered “**Yes**”, please mark the parts of your body where you have been experiencing those pains. If more than one body part is affected, please show which has been the **most important** pain area.” Responders were asked if the pains were caused by serious injury or disease, and infrequent conditions were excluded.

Primary pain conditions (migraine, headaches, growing pains, recurrent abdominal pain) and potentially associated conditions (e.g., history of iron deficiency, anxious depression, multiple sensory sensitivities) were identified by responses to the questionnaires as reported in Champion et al. [14].

Migraine: Diagnostic criteria for migraine with and without aura were as defined by the International Headache Society Classification Subcommittee [32,33], modified for pediatric application [40]. Participants were classified as having a migraine condition if they endorsed all criteria specified on the scale (at least five attacks fulfilling criteria, lasting 4–72 h, at least two of the following: unilateral location, pulsating, moderate/severe pain intensity, aggravation by or causing avoidance of routine physical activity, and during headache, nausea/vomiting, and/or photophobia and phonophobia). There was no separate classification made for individuals who indicated experiencing visual or other aura.

Headache: Participants were classified as having a headache condition if they failed to meet the above criteria for migraine, however, they responded positively to the screening question “Have you/Has your child ever experienced recurring headaches?” and indicated that episodes lasted 4–72 h—untreated or unsuccessfully treated [34].

Growing pains: Diagnostic criteria for growing pains were derived from Peterson [35], Evans and Scutter [36], and applied and modified by Champion et al. [14]. Individuals classified as meeting the criteria for growing pains were those who endorsed four essential criteria (bilateral lower limb pain, onset between 3 and 12 years of age, pain typically occurring at night, and no significant limitation of activity or limping) and denied the presence of exclusion criteria (evidence of orthopedic disorder, abnormalities in any specific tests performed, e.g., X-rays, bone scans).

Recurrent Abdominal Pain: In the same format as above, respondents were asked to indicate whether they/the subject child have had “recurrent abdominal pain (including irritable bowel syndrome)”. Subsequent questions relating to whether this was doctor-diagnosed, age of onset, and resolution were completed if responding positively. For more definitive categorization, the Rome IV criteria for functional gastrointestinal conditions [38] were completed by questionnaire to families in which at least one twin had indicated a positive response to recurrent abdominal pain criteria.

Chronic pain (other than back pain, and not otherwise specified, persisting for at least three months. The screening questions were presented, “Have you had” or “Is it known that your child has (or has had) the following conditions: widespread pain/fibromyalgia or other chronic or recurrent painful disorder”? If yes, participants were asked to specify. The responses were subsequently classified according to the proposed classification for the International Classification of Diseases-11 (ICD-11) of the World Health Organization (WHO) [39]. The classification was completed by one researcher (TB) and confirmed by a senior team member (DC). Where this was not a duplication of another condition assessed in the survey, participants were positively classified in the chronic pain category. Most cases had pain duration greater than three months.

For the classification of restless leg syndrome, we used the criteria from Allen et al. [38]. A positive classification required that participants responded positively to the screening question “Has your child/adolescent ever had a strong urge to move his/her legs, particularly at rest, which may be accompanied by uncomfortable and unpleasant sensations in the legs?”, as well as meeting the essential criteria.

History of iron deficiency was identified by responses to a two-part question (both required): “Have you ever had iron deficiency and was this diagnosed by a doctor”? Serum iron biomarkers such as ferritin do not address the life history of iron deficiency.

Anxious depression was measured by the anxious depression subscale of the Child or Adult Behaviour Checklist from the Achenbach System of Empirically Based Assessment [41]. This measure was age-appropriate and participants were instructed to indicate if each statement was “not true”, “sometimes true”, or “very true” over the past six months. Anxious depression was measured by adding the scores of all the questions, with potential scores ranging from 0 to 36 for adults and 26 for children.

Multiple sensory sensitivity was measured by seven questions which were adapted from the Short Sensory Profile, a 38-item questionnaire based on parent report [42]. This questionnaire investigated the sensitivity of the sense organs. It determined sensitivity to temperature, noise, touch, light, smell, pain, and taste by asking participants how often they had experienced a negative or sensitive reaction to the sensory events [42]. Responses were recorded on a five-point Likert scale. The total score for sensory sensitivity was a continuous variable, ranging from 5 to 35 points.

### 2.4. Statistical Analyses

Summary statistics for LBP (life) and TLBP (current) for twins and family members are presented in Table 1 by sample size and percentage, separately for cases and controls, while mean and standard deviation were used for age. To assess whether there was sampling bias in selecting MZ and DZ twins, we used random effects logistic regression [43] to estimate odds ratios (OR), adjusted for age and gender, if significant. A non-significant OR indicates no sampling bias.

The classical twin design [43] was used to assess genetic and/or environmental factors’ influence on LBP and TLBP. Three methods were used to measure the similarity between MZ and DZ percentages, enabling implications of genetic influence: casewise concordance [44], correlation [45,46], and odds ratio [47,48]. In addition, we also examined relationships between a child and his/her family members for the same trait using ordinary logistic regression. This was done by randomly selecting twins from twin pairs and then performing regression analysis with each of his/her family members with the same trait as the predictor. The same analysis was performed for other twins, and the inverse-variance weighting was used to combine two estimates of the odds ratio.

Random effects logistic regression was used again to assess the relationship between LBP (life) or TLBP (current) as outcome and pain in other body regions as predictors for children, and separately for mothers and fathers. The same method was used for predictors of primary pain conditions and syndromes, history of iron deficiency, anxious depression, and multiple sensory sensitivity.

## 3. Results

Questionnaires were sent to a total of 2479 twin families over two rounds and 651 twin pairs responded (26.3%). Families who responded to the second round with incomplete twin pairs were excluded from the analysis, leaving a total of 175 new families.

Subset analyses showed similar MZ/DZ ratios in both LBP (life) and TLBP (current) samples for those reporting non-serious injury (31.9% of LBP (life)), (39.0% of TLBP (current)) compared with those who did not report injury consistent with no significant effect of injury on heritability assessments, thus, we present analyses for the total LBP (life) and for TLBP (current).

Summary statistics. There were 1307 twin individuals, consisting of 651 complete twin pairs and 5 twin singletons. The twin singletons were excluded in subsequent analyses. Of the complete twin pairs, 275 were MZ (42.2%), 376 DZ (57.8%). There were 638 (49.1%) twin individuals who were male and 662 (50.9%) females. Twins’ ages ranged from 6 to 20 years (M (mean) = 15.3; SD (standard deviation) = 2.42), first siblings’ ages from 4 to 36 years (M = 15.7; SD = 4.61), mothers’ ages from 30 to 64 years (M = 46.3; SD = 4.93) and fathers’ ages from 35 to 78 years (M = 48.7; SD = 5.78). A few twins who were aged 19 or 20 years because of delays in the recruitment and response process and second round recruiting were included in the analyses.

Descriptive statistics for LBP and TLBP for twin pairs, oldest siblings, mothers, and fathers are given in Table 2. There was no significant difference in the proportions of cases between mothers and fathers for LBP (*p* = 0.46) and TLBP (*p* = 0.17), while twins or siblings had lower proportions than mothers or fathers (all *p* < 0.001) for both conditions. MZ twins were slightly older than DZ twins (M(MZ) = 15.7 vs. M(DZ) = 15.0, *p* = 0.0005) but were matched for sex (*p* = 0.51). Adjusting for age, there were no differences between MZ and DZ twins for both types of back pain (*p* (LBP) = 0.430; *p* (TLBP) = 0.367).

The expected association between LBP (life) as predictor and TLBP (current) as outcome was confirmed for twins and siblings combined (OR = 17.7 adjusted for age, *p* < 0.0001), for mothers (OR = 8.84, *p* < 0.0001) and for fathers (OR = 15.7, *p* < 0.0001). There was an association between older age for the twins and siblings combined and both LBP (life) (OR = 1.24, *p* < 0.0001) and TLBP (current) (OR = 1.15, *p* < 0.0001), but no associations between gender and either back pain category.

*Twin analysis*: Casewise concordance for LBP and TLBP (Table 3) showed higher concordance in MZ twins than DZ twins (1-side *p* = 0.008 and 0.006, respectively), suggesting that genetic factors influenced the variability for each of the pain conditions.

Table 4 shows three methods of measuring similarity, i.e., casewise concordance, correlation, and odds ratio methods, consistently higher in MZ than DZ twins.

*Family analysis*: For a randomly selected twin from a twin pair, analysis of association, using ordinary logistic regression, between the twin and his/her co-twin for MZ twins showed that the OR was 9.77 (95% CI = 4.51–21.2) for LBP (life) and 8.99 (95% CI = 3.80–21.2) for TLBP (current), which are similar to those ORs obtained in Table 4.

Associations between randomly selected twins with DZ co-twins, first siblings, mothers, and fathers were significant for LBP (life) (all *p* < 0.01), but with only mother for TLBP (current) (Table 5).

*Associations between back pain and other pain regions*: Associations between LBP (life) and pain in other regions are shown in Table 6. LBP (life) was univariately associated with pain in each other body region, except “head” for children and adolescents (twins and siblings), and each body region for mothers and for fathers. In the multivariate analysis, the significant associations were between LBP (life) and neck pain for children and adolescents, lower limb pain for mothers, and lower limb pain and headaches for fathers.

*Associations between LBP (life) and selected predictors* (primary pain conditions, chronic pain, restless legs syndrome, iron deficiency, anxious depression, and multiple sensory sensitivity) are given in Table 7. In the combined child and adolescent group, LBP (life) was associated with each of the primary pain conditions and syndromes (migraine, non-migraine headache, growing pains, and recurrent abdominal pain) and with other (non-LBP) chronic pain and with anxious depression. Significance was retained in the multivariate analysis for each of the pain conditions.

The significant univariate associations for mothers were migraine, recurrent abdominal pain, other chronic pain conditions, a history of iron deficiency, and multiple sensory sensitivity. Migraine, chronic pain, and iron deficiency remained significant in the multivariate analysis. For fathers, non-migraine headaches, growing pains, other chronic pain conditions, restless legs syndrome, anxious depression, and multiple sensory sensitivity were significant associations in the univariate analysis, while multivariate significance was retained for growing pains, chronic pain conditions, and anxious depression.

*Association analyses for TLBP (current)* similarly showed multiple associations with regional pain conditions and syndromes in children and adolescents and in mothers and fathers (Table 8). For children and adolescents, there were univariate associations with each of the tabulated regions, while neck pain and multiple pains (two or more other regions) were strongly associated and retained in the multivariate analysis. For mothers and fathers, all regional non-back current pain sites were associated with TLBP, especially multiple pain sites, which remained significant in the multivariate analysis.

Associations between TLBP (current) and primary pain conditions, other chronic pain conditions, restless legs syndrome, iron deficiency, anxious depression, and multiple sensory sensitivity are shown in Table 9. In some contrast with similarities between the different age groups for LBP associations, there were more marked contrasts for TLBP. For the children and adolescents (twins and siblings), the univariate associations, each retained in the multivariate analysis, were with migraine, recurrent abdominal pain, other chronic pain conditions, and multiple sensory sensitivity, while there was a univariate association with anxious depression. For mothers, there was a univariate association between TLBP (current) and migraine, chronic pain, and multiple sensory sensitivity and a multivariate association with a history of iron deficiency. For fathers, there was a univariate association with all conditions tested except migraine and iron deficiency, and multivariate significance retained for growing pains, chronic pain conditions, restless legs syndrome, and anxious depression.

## 4. Discussion

In this twin family study, the primary focus (Aim 1) was on genetic influence on and associations with lifetime prevalence of low back pain at least three months duration, i.e., chronic and current thoracolumbar back pain (one-month duration) in twin children and adolescents and their eldest siblings and parents. The analyses were limited in both categories to back pain without substantial trauma or disease, widely referred to as non-specific back pain. Initial or aggravating minor injury was reported in 31.9% of twin responders with LBP (life) and 39.0% of twin responders with TLBP (current). Subset analyses showed no significant influence of reported injury on the determination of genetic influence; therefore, further analyses involved all cases irrespective of history of minor injury. The classification of non-specific low back pain as defined in the European Guidelines on the prevention in low back pain [49] states that non-specific (common) low back pain is not attributable to recognizable, known specific pathology and is chronic when it persists longer than 12 weeks. Severe spinal injury was excluded from the definition. Most cases of non-specific back pain have experienced biomechanical precipitation or aggravation, and that does not exclude such cases from the non-specific classification. Applying ICD-11 terminology and the diagnostic criteria [8] for primary pain stated in the Introduction, the criteria were met for LBP (life) but not for TLBP (current) because those cases were not confirmed to be of three months duration.

Life prevalence of low back pain (LBP (life)) was 12.1% in both the child and adolescent twins and in their oldest siblings. These prevalences for combined children and adolescents were within the wide expected ranges for their respective categories according to epidemiological research [2,5,50]. We were not attempting to assess prevalence estimates, this not being an optimal community sampling method for such an objective.

As expected [1], the prevalence for both back pain categories increased with age, but the prevalence was not greater in females.

The analyses in this twin family study of lifetime prevalence of low back pain of at least three months duration (LBP (life)) in children and adolescents (Aim 1) confirmed parental transmission and showed that parental transmission [24] probably included genetic influence. In view of the evidence for genetic factors in low back pain in adults [18,19,20,21,22,23], this was expected but had not been shown for this young age range. Evidence for parental transmission of TLBP (current) was limited to mothers, i.e., presumed mainly maternal transmission, and the twin analyses indicated genetic influence.

There were multiple associations between both LBP (life) and TLBP (current) and other pain conditions as predictors (Aim 2). Life history of pain in all other regions except headache, increased the risk of LBP (life). Neck and mid (thoracic) back remained significant predictors in the multivariate analyses. Life prevalence of primary pain conditions or syndromes (migraine, non-migraine headaches, growing pains, recurrent abdominal pain and TLBP (current) predicted LBP (life) in the multivariate analyses, as did miscellaneous (non-back) chronic pain. For TLBP (current), a lifetime history of pain in all regions was a significant predictor, especially neck pain and multiple pain sites, which remained significant in the multivariate analysis. The primary pain conditions were less predictive of TLBP (current), being limited to migraine and recurrent abdominal pain. Those two conditions and chronic pain were multivariate predictors of TLBP (current). Mothers and fathers also showed multiple associations between the back pain categories and multiple other pain sites and conditions, although not as consistently with the primary pain conditions as the younger generation.

Have these multiple pain associations with back pain in children and adolescents been previously reported? Frosch et al.’s [1] review barely mentions pain associations. A review of the literature on child and adolescent back pain gives no conception of the multiplicity of pain associations that we have shown (multisite, primary, and miscellaneous chronic). The review on adolescent back pain by O’Sullivan et al. [51] identified multi-dimensional associations, including physical, pathoanatomical, biological, psychosocial, and lifestyle associations, but cited pain associations were limited to neck/shoulder and headache/migraine. Multisite pain, mostly musculoskeletal, is common in adults [52,53] and adolescents, and includes back pain [25,54,55]. There has been less awareness of the extent of associations between the common non-specific or primary pain syndromes of children and adolescents, except by citations of individual conditions such as migraine. The multiplicity of such associations was discussed by von Baeyer and Champion [25] and reported in our twin family study by Donnelly et al. [17].

Why is back pain associated with multiple pain conditions? The risk of back and most chronic and recurrent primary pain conditions increases with age and female sex, and the pain conditions share many physical and lifestyle risks [51]. Multiple pain conditions share genetic factors [56,57,58], and genetic factors influence the female risk [59]. There is a genetic basis to structural grey matter changes in chronic pain [60], and there are neurobiological antecedents of multisite pain in children [61,62]. Kaplan et al. [61] demonstrated augmented activity and connectivity in regions involved in sensorimotor processing and integration as potential vulnerability markers of pain onset in children. Central sensitization, perhaps also nociplastic mechanisms, have been suggested as playing a role [63,64,65]. We have recently shown in four studies in children, adolescents and adults, that a history of iron deficiency is associated, probably causally, with multiple pain conditions [66] (see Appendix A). Other shared risk factors for multiple pain conditions include some bidirectional factors such as impaired sleep [67,68,69], restless legs syndrome [56,70], early childhood adversity [71,72], although bias [73] is acknowledged, and anxiety and depression [74,75,76,77]. Consistent with the anxiety and depression studies cited, both categories of back pain were associated with the measure of anxious depression.

Only LBP (life) was associated in the twins and first siblings with restless legs syndrome (also in fathers). The back pain categories were not associated with a history of iron deficiency, except in the mothers. The measure of multiple sensory sensitivity was multivariately associated with TLBP (current) but not with LBP (life). Both back pain categories were associated with increased sensory sensitivity in both parents.

LBP (life) in adolescents and children shared the following characteristics with the common pediatric primary pain conditions or syndromes (Aim 3) [14]: wide familial and community prevalence; typically, spontaneous onset; multiple risk markers, comorbidities or associations with each other including anxiety and depression; no biomarkers or pathology identifiable by routine clinical or imaging investigations; and genetic influence. The ICD-11 criteria for a chronic primary pain condition were fulfilled. TLBP (current) features were also characteristic of a primary pain condition, but the assessed duration of symptoms, one month, did not enable the fulfilment of the ICD-11 criteria for a chronic primary pain condition.

A key implication for research is that common non-specific or primary back pain, at least in children and adolescents, having so many associations with other common primary pain conditions or syndromes, probably shares causal mechanisms, including shared genes. A genome-wide association study could be used to test this hypothesis. There are implications for clinical practice. A child or adolescent who presents with back pain and has no evidence of disease is at risk of chronicity [1,2,3,4,5,6], and for optimal understanding and management, should be reviewed for family history, other pain conditions, early childhood adversity, stress, anxiety and depression, impaired sleep, restless legs syndrome and iron deficiency, as well as known physical and lifestyle associations.

This study has contributed to the knowledge of back pain in children and adolescents from 3 main viewpoints. Firstly, it is the only study to show that parental transmission significantly includes genetic influence. Secondly, while there are many publications of multisite pain that include back pain in children and adolescents, we have been unable to identify an association study of pediatric back pain with such a broad spectrum of pain conditions. Thirdly, although back pain in adolescence has been mentioned among primary pain syndromes [12,14], this is the first study that has specifically assessed back pain for characteristics determining inclusion among the chronic pediatric primary pain syndromes.

Twin studies are valuable in achieving initial evidence for genetic influence [78,79], and the twin family design was favorable in achieving our aims. Twin studies, however, do have their limitations. The assumption of equal environmental exposure is not necessarily accurate and social contagion is stronger for MZ twins [44]. Our cross-sectional design limits causal conclusions. The diagnoses in our study were based on parent and self-report questionnaires rather than face-to-face medical interviews, this being inevitable in a large-scale epidemiological survey. The diagnostic categorization was thus likely rather than confirmed, lacking some of the checks and balances of a consultation by a specialized clinician. The use of a body map to investigate back pain and other pain sites in adolescents has been evaluated as a reliable and valid method [79]. The response rate, probably influenced by the extent of the questionnaires, was relatively low at 26%. We were unable to control for recall bias. We have not explored environmental factors or gene–environment interactions.

## 5. Conclusions

Common or non-specific LBP (life) in children and adolescents was shown probably to be genetically influenced and was associated with multiple regional and primary pain conditions as well as restless legs syndrome and anxious depression, and fulfilled current ICD-11 criteria for classification as a chronic primary pain condition. TLBP (current) had similar genetic and association characteristics, but a one-month duration was specified and thus was not classifiable as chronic. The genetic influence is consistent with twin and genomic studies in adults and is contrary to the one cited previous twin study in children, which did not show evidence of heritability. Our findings contribute to a better understanding of non-specific/primary back pain, and twin methods combined with molecular genetic studies potentially represent an approach to identify the complex trait variation and shared genes among the primary pain conditions. Focus by treating clinicians on the multiplicity of pain-related and other associations, including anxiety and depression, is expected to be therapeutically advantageous.

## Figures and Tables

**Figure 1 children-10-00375-f001:**
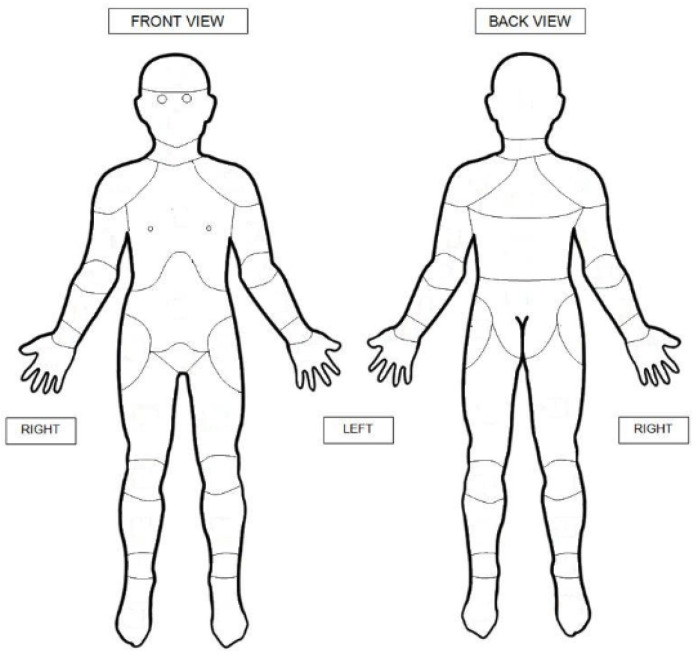
SUPER-KIDZ body map. Copyright permission confirmed: © 2011 Childhood Arthritis and Rheumatology Research Alli-ance (CARRA). www.carragroup.org.

**Table 1 children-10-00375-t001:** Summary of questionnaire content (paper or online).

Category	Method and Content
**Demographic data and zygosity**	Name, twin/sibling/mother/father/guardian, date of birth, gender.
	Zygosity questionnaire [30] completed by carers of twins.
**Back pain**	Questionnaire on pain in the last month, injury causation, past history of back pain, details of current back pain, body map (Figure 1).
**Migraine and headache**	Questionnaire based on diagnostic criteria defined by the International Headache Society classification subcommittee [32,33,34].
**Growing pains**	Modified questionnaire [14,35,36,37].
**Recurrent abdominal pain**	Modified questionnaire [38].
**Chronic pain**	Questionnaire on pain persisting at least three months applying ICD-11 terminology [39].
**Restless legs syndrome**	Criteria from Allen et al. [40].
**History of iron deficiency**	Two-part question. Confirmation by a doctor required.
**Anxious depression**	Child or Adult Behaviour Checklist from the Achenbach System of Empirically Based Assessment [41]
**Multiple sensory sensitivity**	Questionnaire adapted from the Short Sensory Profile [42]

**Table 2 children-10-00375-t002:** Sample size (N) and percentage (%) for cases and non-cases (controls) for LBP and TLBP for twin individuals, mothers, fathers, and oldest siblings.

	Twins N = 1032 (%)		Siblings N = 420 (%)		Mothers N = 640 (%)		Fathers N = 566 (%)	
	Case	Control	Case	Control	Case	Control	Case	Control
**LBP (life)**	158	1144	51	369	269	371	226	340
	(12.1%)	(87.9%)	(12.1%)	(87.9%)	(42.0%)	(58.0%)	(39.9%)	(60.1%)
**TLBP (current)**	124	1171	42	378	170	470	131	435
	(9.6%)	(90.4%)	(10.0%)	(90.0%)	(26.6%)	(73.4%)	(23.1%)	(76.9%)

**Table 3 children-10-00375-t003:** Numbers of concordant and discordant MZ and DZ twin pairs for LBP (life).

		MZ			DZ	
	*N_c_*	*N_d_*	*N* _00_	*N_c_*	*N_d_*	*N* _00_
**LBP**	19	41	215	10	59	307
**TLBP**	13	36	223	5	52	316

*N*_c_ = number of twin pairs concordant with the condition; *N*_d_ = number of twin pairs discordant for the condition; *N*_00_ = number of twin pairs where neither twin has the condition.

**Table 4 children-10-00375-t004:** Twin pair associations for LBP (life) and TLBP (current) using three measures; odds ratio (*OR*), correlation (*ρ*), and casewise concordance (*C*) (N = 651 twin pairs), for MZ and DZ pairs, and statistical significance (*p*) for the difference in association by zygosity.

	MZ		DZ		
	Estimate	95% CI	Estimate	95% CI	*p*
**Odds ratio**	*OR* _MZ_		*OR* _DZ_		
LBP	9.72	(5.63, 16.8)	3.53	(1.99, 6.26)	0.012
TLBP	8.95	(4.87, 16.4)	2.34	(1.12, 4.87)	0.006
**Correlation**	*ρ* _MZ_		*ρ* _DZ_		
LBP	0.39	(0.25, 0.53)	0.17	(0.05, 0.31)	0.014
TLBP	0.34	(0.19, 0.50)	0.09	(0.01, 0.23)	0.010
**Casewise concordance**	*C* _MZ_		*C* _DZ_		
LBP	0.48	(0.35, 0.62)	0.25	(0.12, 0.37)	0.016
TLBP	0.42	(0.26, 0.57)	0.16	(0.04, 0.28)	0.011

**Table 5 children-10-00375-t005:** Univariate association between a randomly selected twin and each (first-degree) family members.

	LBP (Life)				TLBP (Current)			
	N	OR	*p*	95% CI	N	OR	*p*	95% CI
DZ co-twin	376	3.55	**0.002**	(1.58, 8.00)	377	2.37	0.103	(0.84, 6.69)
First sibling	230	3.09	**0.001**	(1.57, 6.08)	230	1.95	0.117	(0.85, 4.47)
Mother	370	3.29	**<0.001**	(1.96, 5.53)	370	2.19	**0.005**	(1.27, 3.76)
Father	334	2.46	**<0.001**	(1.47, 4.10)	334	1.59	0.135	(0.87, 2.91)

See text for MZ co-twin analysis.

**Table 6 children-10-00375-t006:** Associations between pain in other body regions and LBP (life).

	Twins and Siblings (N = 1730)		Mother (N = 646)		Father (N = 569)	
	OR	95% CI	OR	95% CI	OR	95% CI
Multiple sites	5.10 ***	(3.25, 8.00)	4.41 ***	(3.08, 6.33)	5.00 ***	(3.23, 7.74)
Lower limbs	2.24 ***	(1.42, 3.54)	2.58 ***+	(1.81, 3.67)	3.02 ***+	(1.94, 4.71)
Upper limbs	2.87 *	(1.27, 6.51)	2.73 ***	(1.50, 4.97)	2.31 *	(1.12, 4.78)
Abdomen	3.85 ***	(1.69, 8.79)	2.85 *	(1.26, 6.46)	3.43 *	(1.18, 10.0)
Head	1.46	(0.55, 3.87)	2.68 **	(1.34, 5.36)	5.13 **+	(1.65, 16.0)
Neck region	6.56 ***+	(4.04, 10.7)	2.33 ***	(1.59, 3.41)	2.37 ***	(1.50, 3.72)
Middle back	19.8 ***+	(9.05, 43.2)	6.48 ***+	(2.42, 17.3)	7.88 ***+	(3.21, 19.4)

Analysis for twins and first siblings was conducted using random effects logistic regression, adjusted for age, while other analyses were performed using ordinary logistic regression; *p*-value * < 0.05, ** < 0.01, *** < 0.001; + predictors remained significant (*p*-value < 0.05) in multivariate analysis; Multiple sites: pain in two or more non-LBP sites; chest region and whole neck area were excluded because of low numbers.

**Table 7 children-10-00375-t007:** Association between LBP (life) and primary pain conditions, chronic pain, restless legs syndrome, iron deficiency, anxious depression, multiple sensory sensitivity.

	Twins and Siblings (N = 1282)			Mother (N = 487)			Father (N = 430)		
	OR	*p*	95% CI	OR	*p*	95% CI	OR	*p*	95% CI
Migraine	2.16 *	**0.015**	(1.17, 4.02)	1.88 *	**0.016**	(1.13, 3.14)	1.47	0.387	(0.61, 3.55)
Headache	2.39 *	**<0.001**	(1.45, 3.93)	1.08	0.673	(0.74, 1.58)	1.95	**0.008**	(1.19, 3.19)
Growing pains	1.95 *	**0.006**	(1.22, 3.14)	1.46	0.095	(0.94, 2.26)	2.27 *	**0.010**	(1.22, 4.21)
Recurrent abdominal pain	2.94 *	**<0.001**	(1.73, 4.99)	1.71	**0.015**	(1.11, 2.63)	1.59	0.134	(0.87, 2.93)
Chronic pain (other)	3.12 *	**0.001**	(1.58, 6.17)	2.31 *	**<0.001**	(1.48, 3.60)	6.15 *	**<0.001**	(3.18, 11.9)
Restless legs syndrome	1.93	**0.027**	(1.08, 3.45)	1.37	0.130	(0.91, 2.05)	2.60	**0.001**	(1.47, 4.61)
Iron deficiency (history)	1.00	0.287	(0.99, 1.01)	1.60 *	**0.015**	(1.10, 2.33)	1.59	0.575	(0.32, 7.96)
Anxious depression	1.06	**0.040**	(1.00, 1.12)	1.03	0.097	(0.99, 1.07)	1.05 *	**0.009**	(1.01, 1.10)
Multiple sensory sensitivity	0.99	0.701	(0.94, 1.04)	1.04	**0.047**	(1.00, 1.08)	1.07	**0.004**	(1.02, 1.13)

Analysis for twins was conducted using random effects logistic regression, adjusted for age, while ordinary logistic regression was used for mothers and fathers; * predictors remained significant (*p*-value < 0.05) in multivariate analysis; anxious depression and multiple sensory sensitivity are continuous variables.

**Table 8 children-10-00375-t008:** Associations between pain in other body regions and current one-month prevalence of thoracolumbar back pain (TLBP).

	Twins and Siblings (N = 1728)		Mother (N = 646)		Father (N = 569)	
	OR	95% CI	OR	95% CI	OR	95% CI
Multiple sites	22.2 ***+	(13.2, 37.6)	15.2 ***+	(9.98, 23.2)	21.2 ***+	(12.9, 34.9)
Lower limbs	3.89 ***	(2.49, 6.10)	2.67 ***	(1.85, 3.88)	3.12 ***	(1.98, 4.93)
Upper limbs	4.02 ***	(1.82, 8.89)	5.93 ***	(3.24, 10.9)	2.43 *	(1.17, 5.06)
Abdomen	5.52 ***	(2.42, 12.6)	3.62 ***	(1.66, 7.89)	7.94 ***	(2.71, 23.3)
Head	4.71 ***	(2.12, 10.4)	2.77 **	(1.42, 5.41)	3.97 **	(1.50, 10.5)
Neck region	16.8 ***+	(9.59, 29.3)	4.19 ***	(2.82, 6.23)	5.18 ***	(3.23, 8.31)

Analysis for twins and first siblings was conducted using random effects logistic regression, adjusted for age, while for parents was performed using ordinary logistic regression; *p*-value * < 0.05, ** < 0.01, *** < 0.001; + predictors remained significant (*p*-value < 0.05) in multivariate analysis; multiple sites: pain in two or more non-TLBP sites. The chest region was excluded because of low numbers.

**Table 9 children-10-00375-t009:** Association between TLBP (current) and primary pain conditions and syndromes, other chronic pain conditions, restless legs syndrome, iron deficiency, anxious depression, and multiple sensory sensitivity.

	Twins and Siblings (N = 1284)			Mother (N = 487)			Father (N = 430)		
	OR	*p*	95% CI	OR	*p*	95% CI	OR	*p*	95% CI
Migraine	2.66 *	**0.004**	(1.36, 5.20)	1.74	**0.046**	(1.01, 3.00)	1.77	0.231	(0.69, 4.52)
Headache	1.56	0.133	(0.87, 2.78)	1.34	0.178	(0.88, 2.04)	2.24	**0.003**	(1.32, 3.82)
Growing pains	1.67	0.074	(0.95, 2.93)	1.19	0.487	(0.73, 1.95)	2.76 *	**0.002**	(1.46, 5.22)
Recurrent abdominal pain	2.63 *	**0.001**	(1.45, 4.76)	1.30	0.290	(0.80, 2.09)	2.39	**0.008**	(1.26, 4.52)
Chronic pain (other)	3.82 *	**<0.001**	(1.86, 7.84)	2.56 *	**<0.001**	(1.60, 4.08)	3.48 *	**<0.001**	(1.91, 6.34)
Restless legs syndrome	1.60	0.185	(0.80, 3.22)	1.32	0.222	(0.84, 2.07)	3.38 *	**<0.001**	(1.88, 6.08)
Iron deficiency	1.00	0.512	(0.99, 1.01)	1.61 *	**0.026**	(1.06, 2.44)	0.68	0.728	(0.08, 5.91)
Anxious depression	1.10	**0.006**	(1.03, 1.17)	1.02	0.377	(0.98, 1.06)	1.11 *	**<0.001**	(1.06, 1.15)
Multiple sensory sensitivity	1.08 *	**0.002**	(1.03, 1.13)	1.05	**0.046**	(1.00, 1.09)	1.07	**0.009**	(1.02, 1.13)

Analysis for twins was conducted using random effects logistic regression, adjusted for age, while ordinary logistic regression was used for mothers and fathers; * predictors remained significant (*p*-value < 0.05) in multivariate analysis; anxious depression and multiple sensory sensitivity are continuous variables.

## Data Availability

The data presented in this study are available on request from the corresponding author.

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
