# Peer review of "Back Pain without Disease or Substantial Injury in Children and Adolescents: A Twin Family Study Investigating Genetic Influence and Associations"

_children, 2023, doi:10.3390/children10020375_

Round 1
Reviewer 1 Report
The authors presented a study to investigate back pain without disease or substantial injury in children and adolescents. Low back pain (LBP) has been considered that it is common among children and adolescents (Iran J Pediatr. 2011 Sep; 21(3): 259–270.). In children particularly those under 3, LBP is considered as an alarming sign for more serious underlying pathologies. The age of subjects ranges from 6 to 20 in this study, and this study uses questionnaires to collect data. Although, the data presents that the genetic influences on the pain measures. But, I still cannot agree that the genetic issue is a key factor causing the LBP among children and adolescents. I suggest the authors don’t emphasize the gene in the title or conclusions. Other problems are as follows.
1. Please ensure that all abbreviations are defined where they first appear in the manuscripts. For example, what is MZ or DZ?
2. Please explain the novelty of the study. Why is this study important?
3. The manuscript still needs to be checked, for example, some commas that should appear in the sentence are missing.
4. For the citation in the article, “et al” should be corrected as “et al.”.
5. Font size and type are not consistent in the manuscript.
6. Tables 4 and 5 need to be adjusted because they are almost out of the page boundary.
7. In the conclusion, only the important findings of this study are usually described. If there is a need for citing this literature, I recommend to write the sentence in the discussion.

Author Response
The authors presented a study to investigate back pain without disease or substantial injury in children and adolescents. Low back pain (LBP) has been considered that it is common among children and adolescents (Iran J Pediatr. 2011 Sep; 21(3): 259–270.). In children particularly those under 3, LBP is considered as an alarming sign for more serious underlying pathologies. The age of subjects ranges from 6 to 20 in this study, and this study uses questionnaires to collect data. Although, the data presents that the genetic influences on the pain measures. But, I still cannot agree that the genetic issue is a key factor causing the LBP among children and adolescents. I suggest the authors don’t emphasize the gene in the title or conclusions.
Response We thank Reviewer 1 for helpful guidance and suggestions. We are puzzled by the non-agreement with the evidence for genetic influence and do not accept that point of disagreement. The statement was no doubt well meant, perhaps reflecting a personal belief from experience, however no evidence or rational argument was stated. Furthermore, the genetic influence was strong, approved by Prof John Hopper, co-author, who has more than 700 publications mainly on twin studies and genetics, is consistent with the cited evidence for genetic influence on back pain in adults, and was presented cautiously (“probable genetic influence” in the text; “Investigating Genetic Influence” in the Title). The focus was on primary or non-specific back pain, not miscellaneous back pain, and genetic influence is a feature of all pediatric primary pain conditions and syndromes, as we have studied and reported extensively.
Other problems are as follows.
- Please ensure that all abbreviations are defined where they first appear in the manuscripts. For example, what is MZ or DZ?
Response This has been clarified in Abstract and text.
- Please explain the novelty of the study. Why is this study important?
Response We stated, in the Conclusions “Our findings contribute to a better understanding of non-specific/primary back pain, and twin methods combined with molecular genetic studies potentially represent an approach to identify the complex trait variation and shared genes among the primary pain conditions [79]. Focus by treating clinicians on the multiplicity of pain-related and other associations, including anxiety and depression, is expected to be therapeutically advantageous.” Furthermore, we added to the Conclusions “The genetic influence is consistent with twin and genomic studies in adults and is contrary to the cited one previous twin study in children which did not show evidence of heritability.”
- The manuscript still needs to be checked, for example, some commas that should appear in the sentence are missing.
Response Done
- For the citation in the article, “et al” should be corrected as “et al.”.
Response Corrected
- Font size and type are not consistent in the manuscript.
Response Corrected
- Tables 4 and 5 need to be adjusted because they are almost out of the page boundary.
Response Tables adjusted
- In the conclusion, only the important findings of this study are usually described. If there is a need for citing this literature, I recommend to write the sentence in the discussion.
Response The repeat citation [79] was not necessary and was removed from the Conclusions.
The References required minor correction and a new genomic study [23] was added.
Reviewer 2 Report
Children-1974010
Thank you for the opportunity to review this paper.
Manuscript Title: “Back Pain without disease or substantial injury in children and adolescents: a twin familiy study investigating genetic influence and associations”.
Overview: The aims of this article fisrtly is to investigate evidence for genetic factors predicting risk of life prevalence of nonspecific low back pain >3months duration and 1-month current prevalence of thoracolumbar back pain using a study of children, adolescents and their first-degree relatives. Secondly, the study aimed to identify associetions between pain in the back with pain in other regions.
General comments: This is an interesting manuscript that addresses an important area of unmet medical need. Please see my specific comments below for more details.
Specific comments:
Afiliantions: Correct the superscript in author names. I recommend using the template throughout the manuscript,not just the last five pages, for easier review,
- The keywords are absolutely fine, be careful the fifth keyword “family” appears “1” that should not.
3. Introduction: I think they add a lot of information and it becomes difficult to read. It is really necessary to add a paragraph (nº4) explaining "a previous publication in this Journal[14]...", perhaps with reference when talking about pain risk markers in children or adults, it would be enough. In addition, the last paragraph of the intro should make the objective of this study clearer. Please, amend this section.
4. Materials and methods: Be careful is not section 1 but 2. The authors have not reviewed the manuscript, it has many errors of structure and form. The font size is not regular throughout the section. It seems that the manuscript has not been homogeneous writing and a cut and paste has been made.
- Ap. 2.2. Recruitment, participants ans procedure: It would be interesting to add the body map in the manuscript (although it also appears in the supplementary material), but since it is describing the body map, the Figure appears in the text, the easier to understand the explanation.
- Ap. 2.3 Measures and diagnostic categorization: I miss a table that collects all the information that has been collected in the questionnaire: types of pain (LBP life; TLBP(current); Primary pain conditions/potentially associated conditions... and the subscales used in each case. To get an overview of the questionnaire.
5. Results: The authors have not been careful with the shape of the manuscript. We return to another font size different from section 2.3. The tables are not centered. Table 6, 7 and 8 table footer text is not in correct size.
6. Discussion: Suddenly this section appears in the template of the magazine … line 76 Strengths and limitations: Change font size again.
- I consider important your Discussion section where authors has been reflect on the literature published so far on this topic.
Author Response
Overview: The aims of this article fisrtly is to investigate evidence for genetic factors predicting risk of life prevalence of nonspecific low back pain >3months duration and 1-month current prevalence of thoracolumbar back pain using a study of children, adolescents and their first-degree relatives. Secondly, the study aimed to identify associetions between pain in the back with pain in other regions.
General comments: This is an interesting manuscript that addresses an important area of unmet medical need. Please see my specific comments below for more details.
Response Thank you
Specific comments:
Afiliantions: Correct the superscript in author names. I recommend using the template throughout the manuscript,not just the last five pages, for easier review,
Response Corrections made as per template.
2.The keywords are absolutely fine, be careful the fifth keyword “family” appears “1” that should not.
Familial was redundant and was deleted.
- Introduction:I think they add a lot of information and it becomes difficult to read. It is really necessary to add a paragraph (nº4) explaining "a previous publication in this Journal[14]...", perhaps with reference when talking about pain risk markers in children or adults, it would be enough. In addition, the last paragraph of the intro should make the objective of this study clearer. Please, amend this section.
Response We understand that some of the content in the Introduction is detailed and requires the reader to concentrate. However, the concept of primary pain conditions is relatively new, and many readers require explanation. Furthermore, such classification was one of our aims. We could simply refer readers to our referenced publication (14) but thought it would be more coherent for the reader to summarise the features of primary pain here. We have made a minor change to the introduction to paragraph 4: “In a twin family publication on the common primary pain disorders of childhood[14]……”
The aims were summarised in a final paragraph:
The aims, in summary, are:
- To investigate the heritability of chronic low back pain and one-month thoracolumbar back pain in children and adolescents using the classical twin family design;
- To identify associations with the back pain conditions which are most relevant to primary pediatric pain conditions;
- Thereby, in chronic low back pain without disease or substantial injury, to confirm or otherwise that it is classifiable as a chronic primary pain syndrome.
- Materials and methods:Be careful is not section 1 but 2. The authors have not reviewed the manuscript, it has many errors of structure and form. The font size is not regular throughout the section. It seems that the manuscript has not been homogeneous writing and a cut and paste has been made.
Response; Regrettably, you were provided with the original manuscript as mentioned above. We do acknowledge, however, that the originally submitted manuscript had not been adequately checked.
Ap. 2.2. Recruitment, participants ans procedure: It would be interesting to add the body map in the manuscript (although it also appears in the supplementary material), but since it is describing the body map, the Figure appears in the text, the easier to understand the explanation.
Response: The body map is now in the main manuscript as Figure 1.
Ap. 2.3 Measures and diagnostic categorization: I miss a table that collects all the information that has been collected in the questionnaire: types of pain (LBP life; TLBP(current); Primary pain conditions/potentially associated conditions... and the subscales used in each case. To get an overview of the questionnaire.
Response: The questionnaires have been summarised in a new table (Table 1)
Table 1 Summary of questionnaire content (paper or online)
Demographic data: Name, twin/sibling/mother/father/guardian, date of birth, gender.
Back pain questionnaire: pain in the last month, injury causation, past history of back pain, details of current back pain, body map (Figure 1). See Procedure (2.2) and Supplementary Information
Zygosity questionnaire [30] completed by carers of twins: see Supplementary Information.
Primary pain disorders: Questionnaires for migraine [32,33], headache [32,33], growing pains [14], recurrent abdominal pain [37].
Chronic pain: See text 2.3 and reference [38].
Restless legs syndrome: Questionnaire [39].
History of iron deficiency: Two-part question. Confirmation by a doctor required.
Anxious depression: Questionnaire [39].
Multiple sensory sensitivity: Questionnaire (adapted) [40]
- Results:The authors have not been careful with the shape of the manuscript. We return to another font size different from section 2.3. The tables are not centered. Table 6, 7 and 8 table footer text is not in correct size.
Response: again, we acknowledge these technical shortcomings in the initial submission which you received. Further corrections have been made in the re-submitted manuscript.
- Discussion:Suddenly this section appears in the template of the magazine … line 76 Strengths and limitations: Change font size again.
Response: Corrected.
- I consider important your Discussion sectionwhere authors has been reflect on the literature published so far on this topic.
Response: We have re-considered our discussion content, performed multiple literature searches, and added the following penultimate paragraph to the Discussion:
This study has contributed to the knowledge of back pain in children and adolescents from 3 main viewpoints. Firstly, it is the only study to show that parental transmission significantly includes genetic influence. Secondly, while there are many publications of multisite pain including back pain in children and adolescents, we have been unable to identify an association study of paediatric back pain with such a broad spectrum of pain conditions. Thirdly, although back pain in adolescence has been mentioned among primary pain syndromes [12,14], this is the first study which has specifically assessed back pain for characteristics determining inclusion among the chronic pediatric primary pain syndromes.

Reviewer 3 Report
This cross-sectional study investigates the prevalence of primary back pain in children and adolescents and possible genetic predictors using a twin study design.
The questions and methodology are clearly described and meet standard scientific requirements. The results are presented in a comprehensible, detailed and precise manner. The discussion includes the previous relevant preliminary studies on the topic and places them in an appropriate context. The limitations are also adequately described.
Author Response
We thank Reviewer 3 for the supportive review.
Round 2
Reviewer 1 Report
The authors reply to some questions in the response to reviewers, however, some questions are still not replied to appropriately, and they didn't revise the errors in the manuscript.
1) The descriptions about the recruited subject are contradictory. In section 2.1, they mentioned: “In total, we approached 2479 randomly selected twin pairs aged 11-18 years old and their biological parents and siblings.”. In abstract, they claimed: “There were 651 complete twin pairs, aged 6-20 years (response 26%).” Why the age range is different in these two sentences?
2) In the abstract, what is "p’s"? I never see a such way to denote the p value.
3) Did some gene research performed in the study? This is a questionnaire study, wasn't it? Did the authors use bioinformatics methods to analyze the gene data? What specific genes that the authors could confirm the genes were strongly correlated with back pain? In my opinion, this study just can state the heredity is related to back pain. Not gene. Please comment and revise.
4) Don’t cite references in the Conclusions. You just summarize the important findings in this study in the Conclusions. If you want to compare the findings from other studies, please move these sentences to the Discussion.
5) I think the last sentence in the Conclusion is an overstatement. The last sentence is: “Focus by treating clinicians on the multiplicity of pain-related and other associations, including anxiety and depression, is expected to be therapeutically advantageous”. Can the study give a treatment suggestion? I doubt the findings from this study can give a such suggestion. I prefer to suggest the authors give some suggestions to other researchers if they want to perform similar studies.

Author Response
Reviewer 1
Response We thank Reviewer 1 for further comments and recommendations. Please note that Reviewers 2 and 3 accepted the manuscript which you reviewed.
1) The descriptions about the recruited subject are contradictory. In section 2.1, they mentioned: “In total, we approached 2479 randomly selected twin pairs aged 11-18 years old and their biological parents and siblings.”. In abstract, they claimed: “There were 651 complete twin pairs, aged 6-20 years (response 26%).” Why the age range is different in these two sentences?
Response We have corrected the age range in section 2.2 so that it is consistently 11-20. Years.
2) In the abstract, what is "p’s"? I never see a such way to denote the p value.
Response “All p`s<0.02” changed to All p values < 0.02.
3) Did some gene research performed in the study? This is a questionnaire study, wasn't it? Did the authors use bioinformatics methods to analyze the gene data? What specific genes that the authors could confirm the genes were strongly correlated with back pain? In my opinion, this study just can state the heredity is related to back pain. Not gene. Please comment and revise.
Response This was a twin family study by questionnaire and had different aims from genomic research. In part we were rectifying the surprising evidence from the only previous published twin study in pediatric non-specific back pain that it was not heritable. Bioinformatics, as related to genetics and genomics, is a scientific subdiscipline that involves using computer technology to collect, store, analyze and disseminate biological data and information, such as DNA and amino acid sequences or annotations about those sequences. As such, bioinformatics had nothing to do with our research aims. Twin studies are widely accepted as enabling valid evidence for genetic influence, referring commonly to multiple small gene influences. Our senior academic co-author, Professor John Hopper, an international authority in genetics and genomics with more than 700 publications in the field, would not permit us to go beyond evidence-based genetic claims. Perhaps Reviewer 1 might be interested to read explanatory publications about twin research: Hopper, J., Foley, D., White, P., Pollaers, V. Australian Twin Registry: 30 years of progress. Twin Res Hum Genet 2013, 16, 34-42; Hopper, J.L., Bishop, D.T., Easton, D.F. Population-based family studies in genetic epidemiology. Lancet 2005, 366, 1397–1406. Van Dongen, J., Slagboom, P.E., Draisma, H.H., Martin, N.G., Boomsma, D.I. The continuing value of twin studies in the omics era. Nat Rev Genet 2012, 13, 640–653. Craig, J.M., Calais-Ferreira, L., Umstad, M.P., Buchwald, D. The Value of Twins for Health and Medical Research: A Third of a Century of Progress. Twin Res Hum Genet 2020, 23, 8–15.
We modified the fifth paragraph of the Introduction aiming to make more clear the role of the twin analyses in determining genetic influence (We deleted a similar statement from the Methods to avoid repetition): “In this twin family research study, now focusing on back pain, we aimed to investigate familial and genetic influence by applying the classical twin family design. This design stipulates that a greater similarity of MZ twins, who share 100% genes, than the DZ twins who share on average 50% genes, is evidence in favour of familial influence (genetic and/or shared environment), assuming MZ and DZ twins to have equal shared environmental factors. As applied in Champion et al. [14], this design is also applicable to the determination of associations [17] with primary low back and thoracic back pain.”
4) Don’t cite references in the Conclusions. You just summarize the important findings in this study in the Conclusions. If you want to compare the findings from other studies, please move these sentences to the Discussion.
Response Agreed. Those (repeat) references were deleted.
5) I think the last sentence in the Conclusion is an overstatement. The last sentence is: “Focus by treating clinicians on the multiplicity of pain-related and other associations, including anxiety and depression, is expected to be therapeutically advantageous”. Can the study give a treatment suggestion? I doubt the findings from this study can give a such suggestion. I prefer to suggest the authors give some suggestions to other researchers if they want to perform similar studies.
Response That last sentence (Focus by treating clinicians on the multiplicity of pain-related and other associations, including anxiety and depression, is expected to be therapeutically advantageous.) was included on the recommendation by a reviewer that we endeavour to highlight clinical relevance of the study and we believe it to be a true and reasonable statement. However, it is not critical and so has been deleted as requested.
Reviewer 2 Report
Thank you for your effors, congratulations on your manuscript. Now it is.
Author Response
We thank Reviewer 2 for the review and kind words.